# Model Tells You What to Discard: Adaptive KV Cache Compression for LLMs

**Suyu Ge**[1]*, **Yunan Zhang**[1]*, **Liyuan Liu**[2]*, **Minjia Zhang**[2], **Jiawei Han**[1], **Jianfeng Gao**[2]

[1]University of Illinois Urbana-Champaign, [2]Microsoft

{suyuge2,yunanz2,hanj}@illinois.edu
{lucliu,minjiaz,jfgao}@microsoft.com

## Abstract

In this study, we introduce adaptive KV cache compression, a plug-and-play method that reduces the memory footprint of generative inference for Large Language Models (LLMs). Different from the conventional KV cache that retains key and value vectors for all context tokens, we conduct targeted profiling to discern the intrinsic structure of attention modules. Based on the recognized structure, we then construct the KV cache in an adaptive manner: evicting long-range contexts on attention heads emphasizing local contexts, discarding non-special tokens on attention heads centered on special tokens, and only employing the standard KV cache for attention heads that broadly attend to all tokens. Moreover, with the lightweight attention profiling used to guide the construction of the adaptive KV cache, FastGen can be deployed without resource-intensive fine-tuning or retraining. In our experiments across various asks, FastGen demonstrates substantial GPU memory reduction with negligible generation quality loss[2].

## 1 Introduction

Autoregressive language models have attracted extensive attention (OpenAI, 2023; Touvron et al., 2023b). Along with the remarkable success achieved by these models, their computational complexity and GPU memory consumption present significant challenges (Shazeer et al., 2017), underscoring an imperative need for serving these models in an economically feasible manner.

The generative inference of LLMs usually involves using the *KV Cache* mechanism to enhance the generation speed. KV cache stores previously computed Key/Value vectors in attention calculation and reuses those values for the current token generation. As such, it avoids recalculations of previous tokens at each token generation step at the cost of extra memory consumption. Despite being a prominent technique, the memory consumption of KV cache increases rapidly as the model size and generation length increase, drastically increasing the pressure of on-device memory.

When memory usage exceeds GPU capacity, the generative inference of LLMs typically resorts to offloading (Aminabadi et al., 2022; Sheng et al., 2023). However, offloading KV cache to CPU/NVMe adds non-trivial overheads, due to the limited PCIe bandwidth between the GPU and CPU on many devices. Therefore, it is crucial to reduce the memory footprint of KV cache.

Our study starts from the observation that there are abundant structures observed in attention modules (Michel et al., 2019; Voita et al., 2019; Clark et al., 2019; Wang et al., 2020; Child et al., 2019), and as in Figure 1, not all attention modules need to attend to all tokens (Liu et al., 2023b; Zhang et al., 2023; Liu et al., 2023a). Intuitively, harvesting such structures could substantially reduce memory consumption and accelerate text generation.

---

*Authors contributed equally to this research.

[2]We will release code and the compatible CUDA kernel for reproducibility

37th Conference on Neural Information Processing Systems (NeurIPS 2023).

Based on this intuition, we propose FastGen to *accelerate the generative inference by adaptively compressing the KV cache on the fly*. First, we employ an efficient profiling algorithm to recognize the structural patterns for attention modules. Under the guidance of this profiling, we then construct the KV cache for various modules adaptively. Specifically, FastGen recognizes five fundamental attention structures: some attention heads mostly attend to local contexts, for which we construct a KV cache that evicts long-range contexts; some primarily attend to specific tokens/punctuations, for which we create a KV cache that retains only special tokens/punctuations; some have attention maps that are column-wise sparse, for which we discard the least frequently attended tokens; and some broadly attend to all tokens, for which we employ the standard KV cache and store all tokens.

In this way, FastGen is able to compress the KV cache while retaining the original functionality of attention modules, thus does not require any fine-tuning and can be applied in a plug-and-play manner. With a suite of major benchmarks covering generative tasks in math, code, knowledge, and common sense reasoning, we evaluate FastGen on LLaMa (Touvron et al., 2023b) and observe FastGen brings significant improvements over nonadaptive KV cache compression baselines.

---

**Algorithm 1: FastGen–Prompt Encoding.**

**Input:** Feasible Policy Set ($\mathcal{C}$), Prompt
**Output:** Adaptive KV Cache

1 **for** *Attention Head $H_i$ in LLM* **do**
2      $\boldsymbol{K}^i, \boldsymbol{Q}^i, \boldsymbol{V}^i \leftarrow H_i(\text{Prompt})$
3      $\boldsymbol{A}^i \leftarrow \text{softmax}(\boldsymbol{Q}^i \boldsymbol{K}^{i^T})$
4      $\boldsymbol{C}^i \leftarrow \text{apply Equation 1 to } \boldsymbol{A}^i$
     /* C$^i$: optimal policy          */
5      $\boldsymbol{K}^i_{\boldsymbol{C}^i}, \boldsymbol{V}^i_{\boldsymbol{C}^i} \leftarrow f(\boldsymbol{K}^i, \boldsymbol{V}^i, \boldsymbol{C}^i)$
6      $\hat{\boldsymbol{K}}^i, \hat{\boldsymbol{V}}^i \leftarrow \boldsymbol{K}^i_{\boldsymbol{C}^i} \boldsymbol{V}^i_{\boldsymbol{C}^i}$
7 **return** $\{\boldsymbol{C}^i, \hat{\boldsymbol{K}}^i, \hat{\boldsymbol{V}}^i\}$

**Algorithm 2: FastGen–Token Generation.**

**Input:** Adaptive KV cache ($\{\boldsymbol{C}^i, \hat{\boldsymbol{K}}^i, \hat{\boldsymbol{V}}^i\}$)
**Output:** Generated Text

1 $\boldsymbol{z}_0 \leftarrow$ last prompt token
2 **for** $j \in \{1, \cdots, \text{Max Generate Length}\}$ **do**
3      **for** *Attention Head $H_i$ in LLM* **do**
4          $\boldsymbol{K}^i, \boldsymbol{Q}^i, \boldsymbol{V}^i \leftarrow H_i(\boldsymbol{z}_{j-1}, \hat{\boldsymbol{K}}^i, \hat{\boldsymbol{V}}^i)$
5          $\boldsymbol{K}^i_{\boldsymbol{C}^i}, \boldsymbol{V}^i_{\boldsymbol{C}^i} \leftarrow f(\boldsymbol{K}^i, \boldsymbol{V}^i, \boldsymbol{C}^i)$
6          $\hat{\boldsymbol{K}}^i, \hat{\boldsymbol{V}}^i \leftarrow \boldsymbol{K}^i_{\boldsymbol{C}_i}, \boldsymbol{V}^i_{\boldsymbol{C}_i}$
7      $\boldsymbol{z}_j \leftarrow$ sample from LLM prediction
8 **return** $\{\boldsymbol{z}_j\}$

---

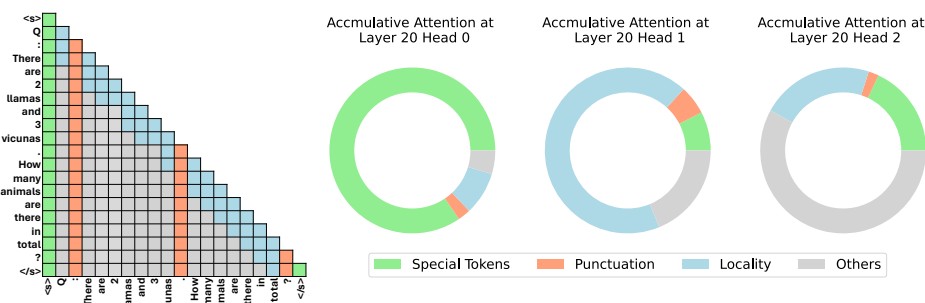

Figure 1: Different attention heads usually have different structures. Left: Four common attention structures. Right: Attention map compositions of three attention heads that are in the same layer.

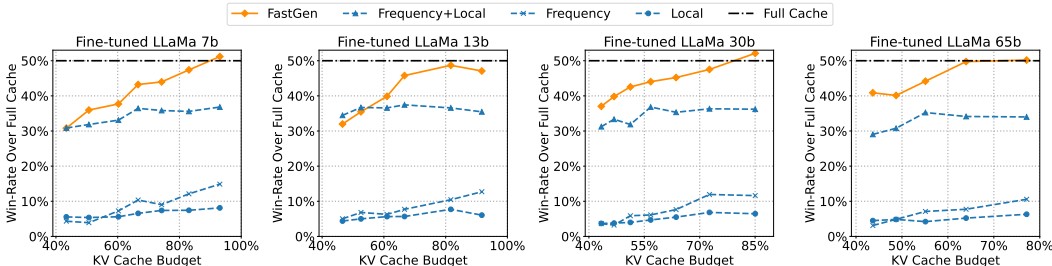

Figure 2: Performance of Adaptive KV Cache (FastGen) and Nonadaptive KV Cache (Frequency, Local, and Frequency+Local; Zhang et al., 2023 and Liu et al., 2023a) on AlpacaEval.

## 2 Related Work

**Token Dropping and KV Cache Compression.** Many efforts have been made to improve the model efficiency for LLMs. For recurrent neural networks, one method is to skip multiple tokens at a given time step (Campos et al., 2017; Seo et al., 2017; Hansen et al., 2019). Since Transformer models quickly attracted lots of attention, Goyal et al. (2020) proposes to eliminate redundant words in BERT (Devlin et al., 2019) based on their attention scores, while Dai et al. (2020) compresses the input sequence by adding pooling layers to the encoding modules of the transformer architecture. Recently, Huang et al. (2022) adds a token selection task to the original BERT model that learns to select performance-crucial tokens, and Kim et al. (2022) designs a learnable threshold to detect unimportant tokens to prune. Meanwhile, many efforts have been made to explore the possibility of compressing the hidden state of tokens rather than explicitly reducing the sequence length (Guan et al., 2022; Sun et al., 2022; Zhou et al., 2020).

Nevertheless, these methods can only be applied to non-autoregressive models and typically require an additional re-training phrase, making them less suitable for large LLMs like ChatGPT and LLaMa. Recognizing this gap, researchers started examining the potential of pruning tokens within the KV cache of autogressive LLMs. Mu et al. (2023) learns to compress the prompts into a few special tokens to reduce memory pressure during caching. However, the token prediction requires training and could be an expensive overhead during inference. Meanwhile, several concurrent methods propose to leverage accumulated attention score as the criteria to identify important tokens in the KV cache (Sheng et al., 2023; Zhang et al., 2023; Liu et al., 2023a). Our work, instead of investigating a specific eviction policy, aims to synergistically coordinate diverse eviction policies, adapting them to align more closely with model-specific attributes.

**Underlying Structure of Attention.** Inspired by the success of Transformer, extensive studies have been conducted to explore the underlying mechanism of different self-attention heads. Voita et al. (2019) analyzed the self-attention heads in BERT (Devlin et al., 2019) using LRF (Bach et al., 2015) and characterized them into interepretable roles, one of which is attending adjacent tokens all the time. Michel et al. (2019) demonstrated that even heads in the same layer could have different impact on the performance while the importance of each head change across tasks. Clark et al. (2019) and Kovaleva et al. (2019) identified the patterns that some heads primarily attend to separator tokens, adjacent tokens and a combination of these. We observe consistent patterns in decoder-only models, despite previous studies are mainly done on encoder models. Our work shares similar spirits with these studies but focus on characterizing the KV cache of different attention heads.

## 3 Adaptive KV Cache Compression

Typical generative inference involves two stages. In **prompt encoding**, LLMs encode prompt inputs. For autoregressive transformers, when generating the $i$-th token, the attention module will reference information from all the preceding $i-1$ tokens. To convey such contextual information, the attention module processes both key and value vectors of the preceding $i-1$ tokens. To circumvent redundant KV vectors computations when generating succeeding tokens, all key and value vectors are retained once encoded, collectively termed as the KV cache. In **token generation**, LLMs conduct incremental token generation. In the process, the associated key and value vectors of newly generated tokens are appended to the KV cache.

As highlighted in Appendix 2, various efforts have sought to compress the KV cache to minimize memory usage and boost generation speed. Yet, these methods often overlook the intricate attention structure in LLMs. As detailed in Appendix 4, attention heads in these models often function distinctively, indicating the need for tailoring compression strategies to each attention head.

In this study, we introduce FastGen: a dual-phase algorithm for crafting an adaptive KV cache.

### 3.1 Model Profiling

FastGenconducts model profiling based on the result of prompt encoding. Specifically, for a compression policy $C$, we mark the corresponding KV cache compression as $K_C, V_C = f(K, V, C)$, where $K_C$ and $V_C$ are the compressed KV cache. Then, for attention map $A = \text{softmax}(QK^T)$,

we pick the optimal policy based on the following objective.

$$C^* = \underset{C \in \mathcal{C}}{\arg\min} \ \text{CacheMemoryCost}(C) \quad \text{s.t.} \quad |A - \text{softmax}(QK_C^T)| \leq 1 - T, \qquad (1)$$

where $\mathcal{C}$ is the set of all feasible compression policies, CacheMemoryCost($C$) is the target KV cache budget of the compression policy $C$, and $T$ is a predefined hyper-parameter representing how much we want the policy to recover $A$. The final profiling algorithm is outlined as Algorithm 1.

Intrinsically, our method assumes that, for all attention heads, the structure of the attention map is stable at different positions, i.e., it is sufficient to use only the encoded prompt for picking the proper compression policy. It is worth mentioning that, existing literature has provided theoretical justification for using only the encoded prompts to capture attention structures for the full contexts (Zhang et al., 2023; Liu et al., 2023a). In our study, we also have provided empirical verification for this assumption (elaborated in Appendix 4).

### 3.2 KV Cache Compression Policies

We observe that a large number of attention heads closely followed certain patterns (elaborated in Appendix 4). While FastGen's design accommodates an expansion to incorporate other strategies, we consider four KV cache compression policies besides the conventional full KV cache ($C_{\text{full}}$):

- $C_{\text{special}}$ only keeps special tokens, e.g., begin-of-sentence token , instruction token [INST].
- $C_{\text{punct.}}$ exclusively preserves punctuation tokens including ".", ":", "?", etc.
- $C_{\text{local}}$ evicts long-range contexts, i.e., once the relative distance between the context token and the current token exceeds a threshold, the KV cache of the context token will be evicted. The length budget of local context over the current sequence length is controlled by a ratio $r_l$.
- $C_{\text{frequent}}$ was proposed in multiple studies (Sheng et al., 2023; Zhang et al., 2023; Liu et al., 2023a). It operates like a least-frequently-used cache by equating the cumulative attention score to a token's frequency. The cache budget over the sequence length is controlled by a ratio $r_f$.

**Naive Strategy Combination.** In practice, we noticed that it is necessary to combine those above-mentioned strategies. However, the total number of the resulting hybrid policy grows at a combinatorial speed. Here, we employed a naive strategy to construct hybrid-policies. Specifically, together with the conventional full KV cache, we consider the following feasible compression policies:

$$\mathcal{C} = \{C_{\text{special}}, C_{\text{special+punct.}}, C_{\text{special+punct.+frequent}}, C_{\text{special+punct.+frequent+local}}, C_{\text{full}}\}, \quad (2)$$

where the sum of two compression strategies is to compute the union of their compressed KV cache. More discussions on the strategy priorities are included in Appendix 5.5. The final algorithm for token generation is outlined as Algorithm 2.

## 4 Diversity and Stability of Attention Structures

In this section, we present empirical verifications on our intuition that KV cache compression can and should be conducted adaptively. Specifically, we aim to first demonstrate that different attention heads typically possess distinct structures; and then, to show that these attention head structures remain relatively consistent. For this purpose, we analyze the attention scores of LLaMa 65B using random samples from GSM8k (Cobbe et al., 2021) as our case study.

### 4.1 Head Distinctive Attention Structure

**Setting.** We apply our proposed profiling method (with a recover threshold of $0.95$) and compute the distribution of profiling results for $\{1, 10, 20, 30, 40, 50, 60, 70, 80\}$ layers. The resulting distribution is visualized in Figure As in Figure 3.

**Observation.** As in Figure 3, attention heads in different layers have vastly different structures. Specifically, for the initial and final layers, they have more attention heads assigned to the full KV cache, indicating more attention heads of these layers broadly attend to all tokens. Meanwhile, for middle layers, the attention map focuses more on special tokens, indicating most attention heads of these layers primarily attend to special tokens (i.e., the accumulated attention score on special

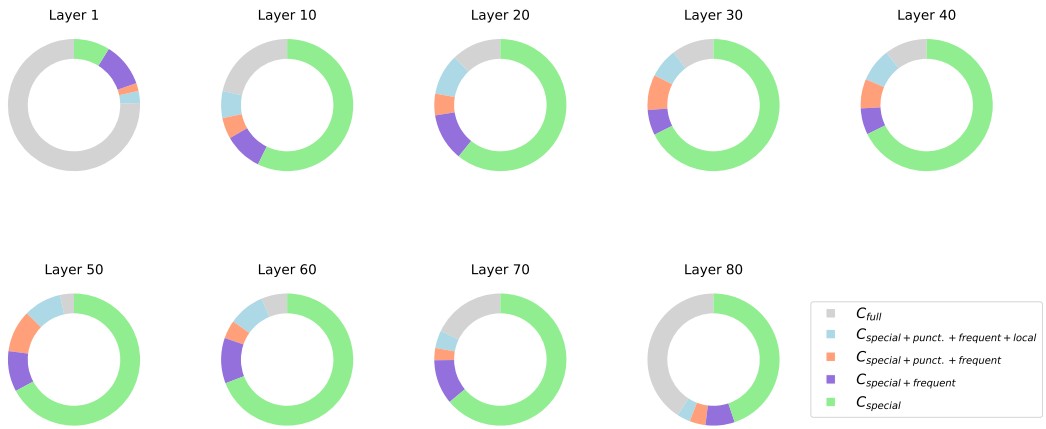

Figure 3: Attention Profiling Result Distribution across different layers.

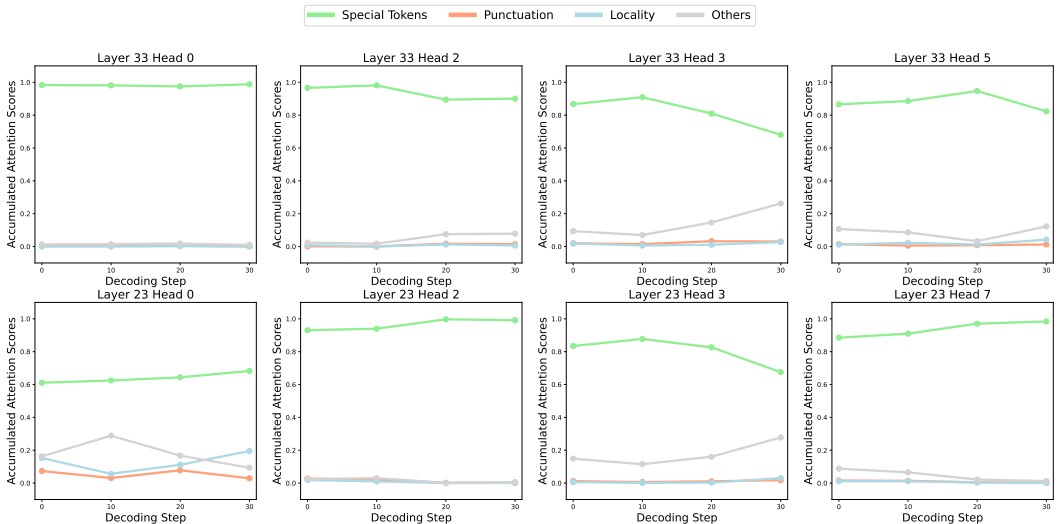

Figure 4: Accumulated Attention Score at 1st (prompt encoding), 10th, 20th, 30th decoding steps.

tokens is higher than $0.95$ for these attention heads). As in Figure 1, we also provide a case study to illustrate the structure of different attention heads in the same layer, which demonstrates that the attention structure differs across different layers and different heads.

Correspondingly, it is suboptimal to follow the same pattern and apply the same KV cache to all layers without adaptation. Also, it could be beneficial to first diagnose the cache of each attention head before deciding how to construct the cache. We'll further discuss the benefit of this diagnose-to-compress strategy in the experiment section.

## 4.2 Profile Tends to Be Consistent in One Sequence

The previous section demonstrates the great potential for constructing an adaptive KV cache in accordance with the structure of different attention heads. Here, we aim to demonstrate that, it is sufficient to leverage only the user-provided prompts and conduct one-shot profiling, as outlined in Section 3. Specifically, we aim to illustrate that user-provided prompts share the same attention structure in the generation process.

**Setting.** Following Figure 1, we compute the accumulated attention score for attention heads in different layers of LLaMa 65B at multiple decoding steps (i.e., 1st, 10th, 20th, 30th). We visualized the resulting accumulated score in Figure 4.

**Observation.** Despite some fluctuations in the exact accumulated attention scores across time steps, the pattern of the attention maps remains relatively stable. For example, Layer 33 Head 0 and Layer 23 Head 2 almost only attend to the special token, while the locality and punctuation plays an important role in Layer 23 Head 0. As to Layer 23 Head 3, more than 10 percent of the attention score is allocated to the Others portion, making it suitable for the uncompressed KV cache $C_{\text{full}}$.

Also, we observe that a large part of attention scores focuses on special tokens in all cases, which matches our intuition for conducting the naive strategy combinations in Section 3.

## 5 Experiment

We conduct comprehensive experiments to demonstrate the effectiveness of FastGen on memory footprint reduction and generation quality preserving. First, we report the trade-off between memory reduction and end-to-end generation quality in Section 5.1, and discuss the compression ratio of FastGen in Section 5.2. Finally, we present ablation studies and discussions in Section 5.5.

### 5.1 Trade-off between performance and memory reduction

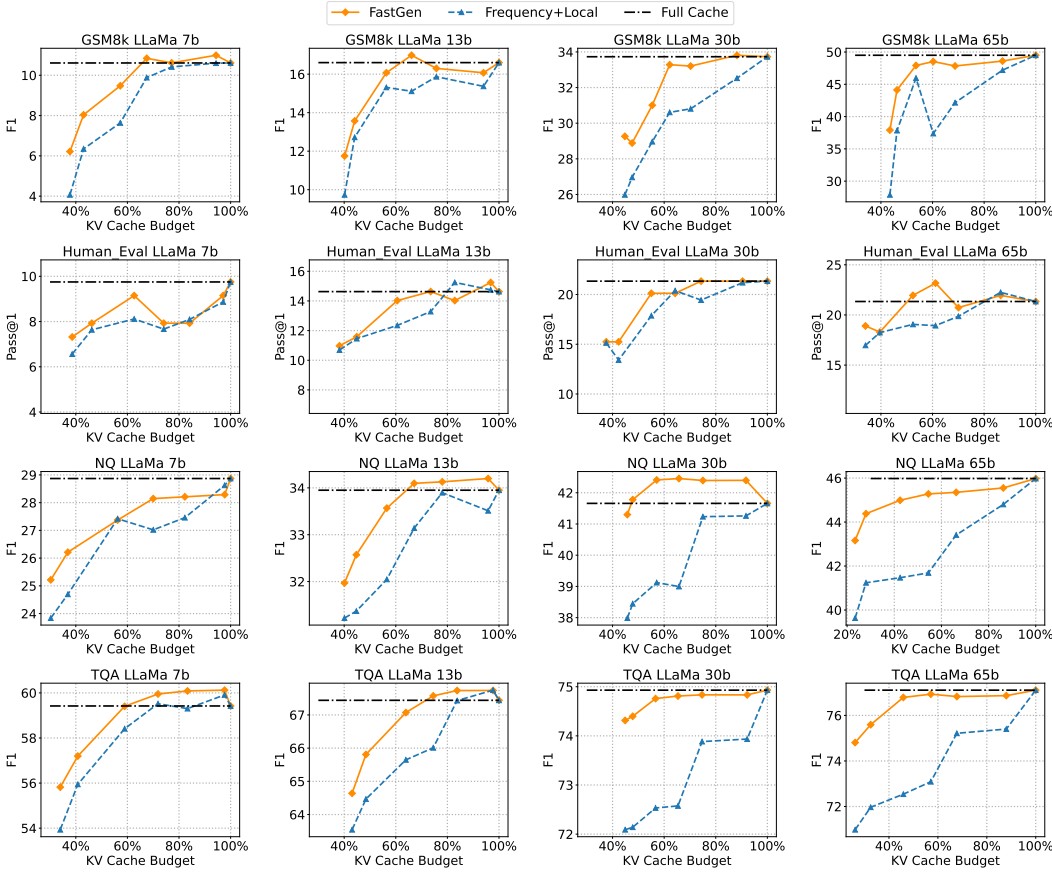

Figure 5: Performance of Adaptive KV Cache (FastGen) and Fixed KV Cache (Frequency+Local; Zhang et al., 2023 and Liu et al., 2023a) of LLaMa on GSM8k, Human Eval, NQ, and TQA.

**Backbones.** We conduct experiments with both LLaMa1 (Touvron et al., 2023a) and its fine-tuned variants, with model sizes ranging from 7B to 65B. For fined-tuned variants, we do not choose the open-sourced ChatLLaMa2 (Touvron et al., 2023b) model due to its grouped-query attention tech-

niques. Instead, we use the original multi-head attention architecture in this study and leave the integration of grouped-query attention to future work. To prepare a comparable instruction-following model for analysis, we fine-tuned the LLaMa1 model with open-sourced instruction-tuning datasets. Specifically, the fine-tuned variants are trained on LIMA[3] data (Zhou et al., 2023) and Open Assistant[4] (Köpf et al., 2023) data.

**Tasks.** We use standard generation tasks to evaluate LLaMa and our fine-tuned LLaMa models. For LLaMa, we choose four different tasks, including Human Eval (Chen et al., 2021), GSM8k (Cobbe et al., 2021), NQ (Kwiatkowski et al., 2019) and TQA (Kembhavi et al., 2017) to evaluate models' abilities on different domains (code, math, question answering and reading comprehension). Note that in the four tasks, each testing sample is in a generative format, where answers are extracted after model generation finishes. This is crucial for a fair comparison on model's generation quality. We evaluate the instruction finetuned LLaMa model on the instruction tuning benchmark AlpacaEval (Li et al., 2023), which consists of 805 question prompts from diverse domains.

**Experiment Setup.** The evaluation of the LLaMa model follows the default setting and evaluation metrics on each benchmark. We calculate F1 scores for GSM8k, NQ and TQA, and use the code execution Pass@1 rate for Human Eval. While evaluating an instruction-tuning model remains challenging, we follow previous work (Zhou et al., 2023; Touvron et al., 2023b) to use GPT4 as an evaluator for pair-wise comparison between two different model generations. For each prompt, we input the FastGen generation and the generation from the same model with Full KV Cache as a pair, and ask GPT4 to judge which one is better. We then calculate the win rate of FastGen over Full Cache. Hypothetically, the win rate of a lossless method should be around 50%. Aside from full-cache models, we also include non-adaptive KV cache methods for comparison. Specifically, we apply $C_{\text{local}}$, $C_{\text{frequent}}$, and $C_{\text{local+frequent}}$ to all attention head without any adaptation, as baselines. It is worth mentioning that $C_{\text{local+frequent}}$ is a very strong baseline as it is identical to the H2O method (Zhang et al., 2023) and the Scissorhands method (Liu et al., 2023a). We set $r_l = 0.3$, $r_f = 0.3$ in FastGen, and only change the recovery ratio $T$ to control the pruned KV cache ratio. For generation, we use nucleus sampling (Holtzman et al., 2019) with temperature T = 0.6, p = 0.9. Experiments are conducted on 8 NVIDIA A100 80GB GPUs.

**Main Results.** In Figure 2 and Figure 5, we present the model quality as a function of KV cache budget increasing from 30% to 100%. For 30B models, FastGen (50% cache compressed) surpasses all non-adaptive KV compression methods (15% cache compressed) . Also, we can see FastGen achieves more KV cache reduction ratio as the model size increases, while preserving the same model quality. For example, given a 45% win rate, FastGencan get as much as 44.9% pruned ratio on LLaMa-65B, compared to 16.9% pruned ratio on LLaMa-7B. In all settings, FastGen shows consistent and significant improvement over non-adaptive compression methods. The results validate the effectiveness of adaptive KV cache compression using FastGen, despite its simplicity.

Table 1: Memory footprint reduction by FastGen. We compared the memory consumption between models with full KV cache, and models compressed by FastGen on fine-tuned LLaMa.

| Model Size | Full | KV Cache FastGen | Pruned ratio | $T$ | Win rate |
|---|---|---|---|---|---|
| 7B | 32Gb | 14Gb | 56.6% | 91% | 30.8% |
| | | 20Gb | 39.8% | 95% | 37.7% |
| | | 27Gb | 16.9% | 98% | 47.4% |
| 13B | 63Gb | 29Gb | 53.4% | 91% | 32.0% |
| | | 38Gb | 39.0% | 95% | 39.9% |
| | | 51Gb | 18.3% | 98% | 48.7% |
| 30B | 158Gb | 69Gb | 56.7% | 93% | 37.0% |
| | | 81Gb | 48.8% | 95% | 42.5% |
| | | 115Gb | 27.4% | 98% | 47.5% |
| 65B | 320Gb | 140Gb | 56.3% | 93% | 40.9% |
| | | 176Gb | 44.9% | 95% | 44.2% |
| | | 205Gb | 36.0% | 98% | 49.8% |

---

[3]https://huggingface.co/datasets/GAIR/lima.
[4]https://huggingface.co/datasets/OpenAssistant/oasst1.

## 5.2 Memory Footprint Reduction Analysis

We report the KV cache memory footprint reduction in Table 1. For all the evaluated 7B-65B models, we evaluate the memory consumption with a fixed batch size of 16, sequence length of 512, and model weights in fp16 format. We observe that FastGen substantially reduces the KV cache memory footprint across all model sizes, with more significant reductions for larger models. Taking a win rate over 45% as no quality regression, FastGen can achieve ∼50% memory reduction in LLaMa-65B, ∼30% in LLaMa-30B, ∼20% in LLaMa-13B and LLaMa-7B.

## 5.3 End-to-end Latency Improvement

Table 2: End-to-end latency comparison on LLaMA7b.

| Batch size | 1 | | | | | 2 | | 8 | | 16 |
|---|---|---|---|---|---|---|---|---|---|---|
| [prompt len, gen len] | [32,512] | [32,2048] | [32,8192] | [32,16384] | [512,32] | [512,512] | [4096,4096] | [512,512] | [4096,4096] | [512,512] |
| Full-cache | 13.35 | 57.37 | 299 | 799.14 | 1.12 | 19.16 | 167.64 | 23.44 | OOM | OOM |
| DeepSpeed | 11.58 | 47.12 | 201.23 | 435.74 | 0.79 | 10.45 | 91.04 | 12.93 | 127.94 | OOM |
| FastGen | 11.21 | 44.6 | 179.43 | 359.83 | 0.73 | 9.71 | 76.93 | 10.57 | 82.16 | OOM |
| Speed-up(%) over Full-cache | 16.03% | 22.30% | 40.00% | 55.00% | 34.80% | 49.30% | 54.10% | 54.90% | - | OOM |
| Speed-up(%) over DeepSpeed | 3.20% | 5.35% | 10.83% | 17.42% | 7.59% | 7.08% | 15.50% | 18.25% | 35.78% | OOM |

To analyze the end-to-end speedup of FastGen, we present the end-to-end latency improvement in Table 2. In the experiment, we record the total duration in seconds, measured from the start of prompt encoding, until the end of generation as the end-to-end latency. For the Full-cache baseline, we use the widely used Hugging Face Accelerate (HF). For FastGen, we implemented a customized kernel to handle the KV cache pruning operation. Specifically, we adapt the kernel from Deepspeed Aminabadi et al. (2022) by adding the KV cache sparsity operation. We include the Deepspeed performance for fair comparison. All methods are tested on the same Nvidia V100 GPUs.

As shown in Table 2, we can observe that FastGen achieves significant end-to-end speed-up across all the generation settings. For the least significant case, Fastgen can have a decent $16.04\%$ latency improvement over the full-cache baseline on a short generation length of 512. In the best cases, we can achieve up to $55.0\%$ latency reduction with Fastgen at a generation length of 16k. We can also observe that the relative speedup is greater with longer generation length. For example, given batch size = 1, FastGen's relative speed-up rises from $16.04\%$ to $55.0\%$, as the generation length grows from 512 to 16k. When comparing FastGen to DeepSpeed, we can still observe significant speed-up that gets bigger with batch size and generation length. Considering DeepSpeed is a full-stack optimized inference system, there is still much room to further improvement FastGen by polishing the sparsity kernel. We leave this unique research and engineering challenge to future works.

## 5.4 Profiling Cost

Table 3: Profiling time of LLaMA65b. The Overall Generation Duration is measured from the start of decoding to the end of the generation length. The Profiling Duration is measured from the start of the decoding until Fastgen finishes the policy search.

| Generation Length | Overall Generation Duration (s) | Profiling Duration (s) | Decoding Time Per Token (s) | Profiling/Overall (%) |
|---|---|---|---|---|
| 128 | 30.98 | 0.11 | 0.1 | 0.35% |
| 256 | 50.1 | 0.11 | 0.1 | 0.21% |
| 512 | 94.98 | 0.11 | 0.1 | 0.12% |
| 1024 | 157.43 | 0.11 | 0.1 | 0.07% |

To better understand the overhead of the profiling step, we compare the profiling time with the total generation time across different generation lengths. We present the result in Table 3.

We can observe that the profiling time only accounts for a very small percentage of the total generation duration, up to $0.35\%$ in our tested cases. Also, the overhead decreases as the generation length increases, dropping to $0.07\%$ when the generation length comes to 1024.

In terms of extra memory usage, it's mainly introduced by one of the compression strategies, $C_{\text{frequent}}$, which needs to store an extra cumulative sum of attention scores for each attention head. To provide a detailed analysis, for each layer, the dimension of the KV cache is

Table 4: Complementary effects of each policy. We display the win rate of each method over full cache setting. We evaluate the fine-tuned LLaMa 65B on AlpacaEval with the same parameters.

| Feasible Policy Set | Pruned KV Ratio | Win Rate |
|---|---|---|
| $\mathcal{C}$ | 36.04% | 49.75% |
| $\{C_{\text{punct.}}, C_{\text{punct.+frequent}}, C_{\text{punct.+frequent+local}}, C_{\text{full}}\}$ | 31.16% | 47.64% |
| $\{C_{\text{special}}, C_{\text{special+frequent}}, C_{\text{special+frequent+local}}, C_{\text{full}}\}$ | 34.23% | 49.56% |
| $\{C_{\text{special}}, C_{\text{special+punct.}}, C_{\text{special+punct.+frequent}}, C_{\text{full}}\}$ | 30.18% | 49.06% |
| $\{C_{\text{special}}, C_{\text{special+punct.}}, C_{\text{special+punct.+local}}, C_{\text{full}}\}$ | 21.26% | 46.08% |

Table 5: Policy order ablation on fine-tuned LLaMa 65B with AlpacaEval.

| Cache Order | Pruned KV Ratio | Win Rate |
|---|---|---|
| $C_{\text{special}} \to C_{\text{punct.}} \to C_{\text{frequent}} \to C_{\text{local}}$ | 36.04% | 49.75% |
| $C_{\text{special}} \to C_{\text{frequent}} \to C_{\text{local}} \to C_{\text{punct.}}$ | 36.40% | 47.64% |

(`batch_size, num_of_head, sequence_len, hidden_dimension`), while the dimension of extra memory for the cumulative attention scores is (`batch_size, num_of_head, equence_len`). Considering `hidden_dimension` = 128 for all model sizes, the memory overhead is 1/128=0.78% compared to storing KV cache only, which is a negligible cost.

## 5.5 Ablations

For all the ablations, we use a fixed targeted recovery ratio T = 0.98.

**How one policy affect all the other policies?** We study the complementary effects of each policy on the combination of all other policies in our framework. We examine changes in pruned KV cache and win rate while fixing the targeted recovery ratio $T$. We take the full policy set as our control set $\mathcal{C}$. For each ablation, we remove one of the policies from all policy combination in $\mathcal{C}$. We summarized the results in Table 4, which suggests the $C_{\text{frequent}}$, and the $C_{\text{special}}$ are the most important policies. Removing them will incur a 3.67% and a 2.11% win rate drop respectively. We can also observe from the pruned cache ratio that $C_{\text{frequent}}$ and $C_{\text{local}}$ reduce more KV caches than the others. However, their standalone non-adaptive deployment yields suboptimal performance, as depicted in Figure 2, further verifying the importance of adapting different compression policies.

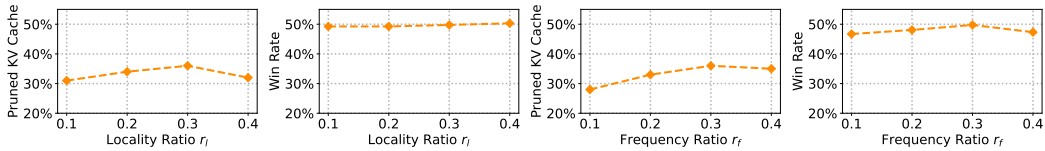

Figure 6: Hyper-parameter ablation on fine-tuned LLaMa 65B with AlpacaEval.

**Which policy should we add first (and last)?** As in Section **??**, we use a greed method to construct adaptive KV cache. Here, we examine how the order of introducing each policy affects the performance. Similar to the previous study, we fix the targeted recovery ratio to 0.98, and keep allocating cache budget until the constructed cache hit the recovery ratio. For simplicity, we make every examined order opt-in the $C_{\text{special}}$ first, as it's typically the most important tokens and of super-low memory cost, as suggested in Figure 1. We summarize the results in Table 5. Our current order (as in Equation 2) achieves the highest win-rates and the highest pruned ratios. Meanwhile, using alternative orders leads to a different trade-off between KV cache compression and generation quality. For example, using $C_{\text{frequent}} \to C_{\text{local}} \to C_{\text{punct.}}$ leads to an improved KV cache compression ratio at the cost of generation quality.

**Sensitivity Study.** We analyze the sensitivity of selecting different hyper-parameters for FastGen, as illustrated in Figure 6. We observe that altering these hyper-parameters does not have a visible impact on the generation quality, as the model maintains a winrate over 45% in all situations. Meanwhile, it leads to a relative large change on the compression ratio. For example, changing the ratio for the frequency policy from 0.3 to 0.1 leads to +10% more KV cache. In our experiments, we set the ratio to 0.3 for both $r_l$ and $r_f$.

## 6 Conclusion

In this work, we introduce FastGen, a novel method that improves the efficiency of large language models through lightweight attention profiling and adaptive key-value caching. Our approach enables more efficient deployment across diverse models and datasets without loss of generation quality. Further work includes combining FastGen with other optimization techniques like quantization and distillation. Meanwhile, another direction could be accommodating other efficient attention architecture, such as grouped-query attention. We hope our initial results will spur further developments in efficient and deployable LLM technologies.

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
