# OpenReview forum: "Model Tells You What to Discard: Adaptive KV Cache Compression for LLMs"
_NeurIPS.cc/2023/Workshop/WANT — WANT@NeurIPS 2023 Poster_

### Official Review · Reviewer_n8Gu · 2023-10-22
**The KV cache can be compressed by identifying the attention score structure of the prompt encoding**

**Confidence:** 4

**Review:**

The paper addresses a really timely issue: how to alleviate the memory footprint of the KV cache during LLM inference. I find their strategy quite original: identifying five common attention structures, each of them allowing to save the K and V of specific tokens instead of the full sequence length.
The authors based their approach on two assumptions: 1) different layers and heads need different attention structures and 2) the attention structure identified during the prompt encoding is maintained during the token generation. I have some caveats about how the authors justify the validity of the assumptions.
The authors show results using GPT-4 as an evaluator, for model generations with full KV cache and their compressed KV cache (plus other non-adaptive methods in the literature). This is fine but I miss other more standard comparisons, like MMLU or WinoGrande.

Pros:
- Instead of more naive approaches to compress the KV cache like sparse sampling or non-adaptive strategies, their method identifies the particular attention structure of a prompt (out of five possible structures) and applies it to compress the KV cache.
- Contrary to other methods in the literature that apply the same structure to all heads and layers, their authors show the need to be more granular and apply different structures per head and layer.
- Their method surpasses previous non-adaptive methods in the literature and matches the results using full KV caching, when using Llama 1 generation and comparing the generation quality with GPT-4.

Cons:
- For the assumption of "Head Distinctive Attention Structure" in section 4.1, I'm missing more detail about 1) how the attention structure is identified in each head and 2) a comparison with more datasets that are not GSM8K (which is based on math word problems with fixed structure). I'm not sure if the claim of "for the middle layers, the attention map focuses more on special tokens" would hold for other datasets.
- For the assumption of "Profile Tends to be Consistent in One Sequence" in section 4.2, I have two issues with their analysis:
1) the authors only show two intermediate layers (33 and 23), and a few heads (0, 2, 3 and 5). Could the authors add a summarized view of all the layers and heads, as it is done in section 4.1? Maybe a metric like "average derivative" is enough for the visualization.
2) the authors only generate 30 tokens, which I find not long enough compared to a normal 1024 generation length. In addition, some plots (see the ones for head 3) present a significant trend even after 30 tokens.
- In line 257, it says that there's a "~50% memory reduction in Llama-65B", but in Table 1 I see that even with 44.9% reduction it is not reaching the 45% win rate. Are the authors rounding up that 44.9% to 50%? It seems quite a high jump in my opinion. Same with the numbers for the other model sizes.
- I didn't find a mention about "sequence length" or number of generated tokens in the experiments of section 5.1. I believe it is important to mention it, specially given the assumption taken in section 4.2. Is the sequence length reported in line 254 (i.e. 512) the one used in the experiments in Figures 2 and 5?

Questions to clarify:
- In the line 11 of the abstract, the authors mention "FastGen" for the first time. At this point, it is not clear what "FastGen" refers to.
- In line 54, the citation is incorrect: the citation should refer to Llama 1, not Llama 2. Later in the experiments of section 5 this is correctly stated.
- In line 94, the authors use the term "Prompt encoding". I've usually seen this part referred as "prefill" in the literature.
- Typo in line 175: remove "in Figure As".
- Typo in line 195: change "visualized" for "visualize".

---

### Official Review · Reviewer_5RX7 · 2023-10-24
**Concise, clear, and well-supported paper on decreasing Transformer inference memory requirements via dynamic caching**

**Confidence:** 5

**Review:**

# Summary

The paper proposes a model-adaptive key-value (KV) cache compression
for autoregressive Transformer models. The key contributions include
identifying notable patterns for compression and combining and
selecting compression schemes for ideal memory/quality balance. The
compression scheme is dynamically selected on the fly for each input
prompt and re-used for autoregressive generation.

The scheme is evaluated on both small and large scale and indicates
strong scaling abilities. Ablations were conducted to optimize the
scheme and understand the significances of its hyperparameters.

The publication is strong and the scientific community would benefit
from its inclusion at NeurIPS. However, when taking the title of the
workshop literally (&ldquo;Workshop on Advancing Neural Network Training&rdquo;),
I am not sure the topic matches the intended audience. The publication
is not about advancing training, but inference/deployment. I would
personally consider accepting the paper nonetheless due to its
intuitive, well-performing, and easily applicable improvements based
on modern understanding of Transformers.

Edit: This technique could be used to improve scaling of the reinforcement learning, which follows pre-training, since it requires autoregressive generation.

# Strengths

-   Bases its intuitive but useful idea on lots of empirical evidence
    from the literature. This evidence has been confirmed independently
    in multiple publications and seems rock solid.
-   Manages to improve upon existing KV cache compression methods in
    memory and quality with simple additions to existing methods.
-   Introduces an adaptive KV caching scheme.
-   Introduces new KV caching policies from combinations of existing
    policies.
-   Method, experiments, and results are clearly and concisely
    communicated and easily understandable. (Partly due to the beautiful
    simplicity inherent to the method.)

# Weaknesses

-   Mention of and values of latency effects from initial profiling are
    missing; also consider it in relation to policy set.
-   Would appreciate at least some implementation details; maybe briefly
    mention special CUDA kernels required due to introduced sparsity in
    the main text? Additionally, since this is a highly practically
    relevant paper for deployment, mention ease of integration into
    existing systems?
-   Throughput changes? At least, mention potential effects on
    throughput (higher ops/byte ratio, &#x2026;).
-   Ablation experiment selection seemed arbitrary; the selection should
    at least have been explained, or ideally more exhaustive ablations
    ran.
-   Ablations were only conducted on one benchmark (AlpacaEval).
-   Fine-tuning trainings and following results are not reproducible
    without further details.

# Additional Suggestions/Feedback

-   Additional reference for attention sinks at special and punctuation
    tokens: <https://arxiv.org/abs/2306.12929>
-   Explain how grouped-query attention changes paper&rsquo;s results. Do we
    need new empirical results for these models? Maybe existing results
    are reusable (maybe there&rsquo;s a reference for this?)?
-   Option of different recovery ratios $T$ for different heads can be
    explored.
-   Future of the approach if QuietAttention
    (<https://www.evanmiller.org/attention-is-off-by-one.html>) becomes
    more widely used and attention sinks (special tokens/punctuation)
    become less common?
-   The pie circle diagrams are very stylish.

# Ordered Notes

Since this review became more detailed than intended, I tagged the
less important language-related notes with &ldquo;(LANG)&rdquo; and style-related
notes with &ldquo;(STYLE)&rdquo;. You can filter these out if you prefer to focus
on the more important, scientifically relevant notes.

Note that I am not a native English speaker, so please re-check my
language-related suggestions.

## General:

When the text refers to Llama&nbsp;2, it is also called &ldquo;LLaMa&rdquo;, which was
one of the multiple names of the first model in the series. Ideally,
it would be referred to by the &ldquo;normalized&rdquo; name introduced in
Llama&nbsp;2, both in the text and figures. I understand if this is too
much work for little gain.

Additionally, please explicitly mention which version you are
comparing to at least once in the beginning (like it is done in
Section 5), if not everywhere. This makes for easier understanding
without having to check the reference.

## General: (STYLE)

The light green color used for &ldquo;special tokens&rdquo; will probably become
hard to read when projecting the line plots (Figure 4) using a beamer.

## General: (STYLE)

The soft and lightly contrasted green/orange/blue/grey style does not
match the stark and heavily contrasted blue/orange matplotlib default
style (used in Figures 2, 5, and 6).

## General: (LANG)

&ldquo;Key&rdquo;, &ldquo;value&rdquo;, &ldquo;transformer&rdquo;, and &ldquo;b&rdquo; (for billion) are arbitrarily
capitalized; ideally this would be adjusted so they have the same
capitalization everywhere.

## Page 1, line 27: (LANG)

I would change

> the generative inference of LLMs typically resort to

to something like

> a typical resort for GPU-enabled generative inference of LLMs is

It&rsquo;s not the inference that resorts to offloading; it&rsquo;s the
programmers when trying to optimize.

## Page 1, line 32:

Maybe also mention potential quality loss due to re-training/fine-tuning.

## Page 2: (STYLE)

Put algorithm and figures at top of page so the flow of reading is not
interrupted.

## Page 2, Figure 1 and 2: (STYLE)

Please move graphics up to increase caption readability. If you need
vertical space, consider moving the diagram of Figure 3 (page 5)
closer toward its caption; there&rsquo;s a lot of space between the diagram
and caption in Figure 3.

## Page 2, lines 46–48: TODO

Not clear how specific token/punctuation differs from column-wise sparsity

## Page 3, line 53: (LANG)

affordable for → afforded by

## Page 3, line 54:

Llama&nbsp;2 is cited here, but the experiments in Section 5 only use
Llama&nbsp;1. Please clearly indicate which version of Llama is used in
which section or adapt the citation if only Llama&nbsp;1 was used in the
experimental portions of the paper.

## Page 3, lines 57–58:

The same finding of always surpassing the baseline could also be said
for the 65B model, if I read Figure 2 correctly. I would mention this
as well to emphasize the scaling abilities of FastGen.

## Page 3, lines 64–65:

(With regard to line 71.)

[Hourglass Transformer](<https://arxiv.org/abs/2110.13711>) uses pooling
similar to the Funnel-Transformer, but in an autoregressive
architecture. However, the argument regarding re-training still
stands.

## Page 3, line 84: (LANG)

interepretable → interpretable

## Page 3, line 86: (LANG)

head change across → head changes across

## Page 3, lines 87–88:

> and a combination of these

Would &ldquo;and/or a combination of these&rdquo; be more correct?

## Page 3, line 89: (LANG)

are mainly done on → mainly considering

## Page 3, line 90: (LANG)

focus → focuses

## Page 3, line 98: (LANG)

vectors → vector

## Page 3, line 100: (LANG)

finished → is finished

## Page 3, line 102: (LANG)

being generated → is generated

## Page 4, line 103: (LANG)

experiences a linear surge → grows linearly

## Page 4, line 120: (LANG)

by a recover ratio → with a recovery ratio

## Page 4, lines 125–126:

> FastGen is able to recover over 95% percent of attention map with over 40% compression ratio

should be something like

> FastGen is able to recover attention maps with recovery ratios of over 95% for compression ratios of over 40%

## Page 4, lines 147–149:

I think here a short formula would be really beneficial.

## Page 4, lines 139–146:

Please add references for the other 3 policies. (See also additional
reference at top of the review.)

## Page 5, lines 156–164:

Pleas explain why only successively growing combinations were
considered. For example, why is there not a separate entry
$C_{special+punct.+local}$ that excludes the &ldquo;frequent&rdquo; policy but
includes the &ldquo;local&rdquo; policy?

## Page 5, line 174:

> {1, [&#x2026;], 80} layers

should be

> layers {1, [&#x2026;], 80}

## Page 5, line 175: (LANG)

in Figure As in Figure 3 → in Figure 3

## Page 6, line 194:

Maybe explicitly mention that attention scores are accumulated over
all tokens that each policy considers.

## Page 6, line 204: (LANG)

Experiment → Experiments

## Page 6, line 206: (LANG)

preserving → preservation

## Page 7, lines 211–212:

The same argument for not using Llama&nbsp;2 fine-tuned variants (i.e., for
not using Llama&nbsp;2-Chat) could/should be made for the non-fine-tuned
variants.

## Page 7, line 224: (STYLE)

Human Eval → HumanEval
(Same for later mentions.)

## Page 7, line 224: (LANG)

instruction finetuned → instruction-finetuned

## Page 7, line 224: (LANG)

on instruction tuning → on the instruction-following

## Page 7, line 229: (LANG)

instruction-tuning model → instruction-tuned model
or
instruction-tuning model → instruction-following model

## Page 8, line 237: (STYLE)

H20 → H$_2$O

## Page 8, line 237: (LANG)

pruned KV cache ratio → KV cache pruning ratio

## Page 8, line 244: (LANG)

(15% cache compressed) . → (15% cache compressed).

## Page 8, line 246:

given a 45% win rate → achieving a 45% win rate

## Page 8, line 246: (LANG)

FastGencan → FastGen can

## Page 8, line 248: (LANG)

despite they being → despite them being a

## Page 8, Table 1:

> models compressed by FastGen

should be

> models with caches compressed by FastGen

After all, your technique does not compress the model itself.

## Page 8, line 253: (STYLE)

7B-65B → 7B&#x2013;65B

Using two hyphens makes the dash a bit longer in LaTeX, which looks
nicer for ranges.

## Page 8, lines 256–257:

It&rsquo;s hard to argue that a 5% win rate loss is &ldquo;no quality regression&rdquo;;
this should be phrased differently.

## Page 8, lines 256–257: (STYLE)

~ → $&asymp;$
(3 occurrences)

## Page 8, line 261: (LANG)

watch changes in pruned KV cache → monitor changes in the KV cache pruning ratio

## Page 9, lines 267–268, Table 3 and Figure 6: (STYLE)

There are two lines of main text between Table 3 and Figure 6; please
move the figure right below the table to increase the flow of reading.

## Page 9, lines 269–278:

Please add an explanation why just these two orders chosen or add
additional results. For example, another valid order to try would have
been $C_{special} → C_{punct.} → C_{local} → C_{frequent}$.

## Page 9, lines 279–284:

Just for completeness sake, I&rsquo;d appreciate explaining why even with
higher ratios, you will not always get higher win rates/cache sizes.
Something like &ldquo;Small changes due to an improved reconstruction may
lead to different outcomes, which&nbsp;– even considering they are a better
reconstruction&nbsp;– may be less preferred by GPT-4. Similarly, even if
more data is stored according to the ratio, the cache required may be
smaller due to the differences in outcomes&rdquo;.

## Page 9, lines 282–284:

According to the figure, the cache does not increase by more than 10%
when going from frequency ratio 0.3 to 0.1; it&rsquo;s slightly less than
10%.

---

### Official Review · Reviewer_o7ip · 2023-10-24
**Novel approach for KV-cache compression with meaningful performance gain**

**Confidence:** 3

**Review:**

Authors propose FastGen, a novel approach for KV-cache memory footprint reduction based on profiling attention head structures and deriving efficient compression policies from it. The idea is clear, the paper is well-written and the experiments are compelling. An important strength of the proposed approach is plug-and-play design which does not require any re-training.

Questions for the authors:
1. While FastGen is a clear improvement over Fixed KC Cache, it would be great to include comparison to other approaches that compress KV-cache memory footprint such as quantization, distillation, and grouped-query attention mentioned in the Conclusion. By construction, FastGen is orthogonal to these approaches and adapting it to work together will go beyond the score of the paper, however, it would be great to have the standalone comparison, such as in Table 1.

Minor issues and typos:
1. Line 11 — FastGen is mention without being introduced
2. Line 246 — FastGencan typo

---

### Meta-Review · Area_Chair_iaH6 · 2023-10-25

**Recommendation:** Accept (Poster)
**Confidence:** 4

**Metareview:**

Pros:
Novel, adaptive compression ratio of the KV cache, which achieves meaningful compression ratios. Overall the paper is well-explained and the effectiveness of the method is examined well.

Cons:
Some things are still unclear in the manuscript and reviewers had open questions. I am certain the authors will have a look at the reviewer’s remarks and  address these open issues.

---

### Decision · Program_Chairs · 2023-10-28

**Decision:**

Accept (Poster)

**Comment:**

We thank the authors for their time and contribution to WANT and we are pleased to share that after the reviewing process the paper has been accepted. Congratulations! We encourage the authors to consider reviewers' feedback for the improvement of the camera-ready version. We hope to see you in person at the workshop and brainstorm on efficient training research together!